# USING OBJECT-FOCUSED IMAGES AS AN IMAGE AUGMENTATION TECHNIQUE TO IMPROVE THE ACCURACY OF IMAGE-CLASSIFICATION MODELS WHEN VERY LIMITED DATA SETS ARE AVAILABLE

## ABSTRACT

Today, many of the machine learning models are extremely data hungry. On the other hand, the accuracy of the algorithms used is very often affected by the amount of the training data available, which is, unfortunately, rarely abundant. Fortunately, image augmentation is one of the very powerful techniques that can be used by computer-vision engineers to expand their existing image data sets. This paper presents an innovative way for creating a variation of existing images and introduces the idea of using an Object-Focused Image (OFI). This is when an image includes only the labeled object and everything else is made transparent. The objective of OFI method is to expand the existing image data set and hence improve the accuracy of the model used to classify images. This paper also elaborates on the OFI approach and compares the accuracy of five different models with the same network design and settings but with different content of the training data set. The experiments presented in this paper show that using OFIs along with the original images can lead to an increase in the validation accuracy of the used model. In fact, when the OFI technique is used, the number of the images supplied nearly doubles.

## 1 INTRODUCTION

Nowadays, Convolutional Neural Networks (CNNs) are among the most common tools used for image classification. For machine learning (ML) problems such as image classification, the size of the training image set is very important to build high-accuracy classifiers. As the popularity of CNNs grows, so does the interest in data augmentation. Although augmented data sets are sometimes artificial, they are very similar to the original data sets. Thus, augmentation can make a network learn more useful representations because of the increase of training data.

In this paper, the researchers propose a new method to produce new images from existing ones and therefore augment data. Several experiments were conducted to validate that the model could benefit further from using the mixed data set of old and new images.

This paper compares five models with the same design and settings. The only difference is the set of supplied images. The first model uses the original 2,000 images of dogs and cats. The second and the third use the OFI version of those 2,000 images. The fourth and the fifth use all the images: the original images as well as the OFI images. There are two methods to obtain the OFIs. The automated method uses an Application Programming Interface (API) to remove the background of the image and leave only the labeled object (a cat or a dog) in the foreground. The OFIs produced by the first method are called automatic OFIs. The other method is executed manually: every image is edited by a human expert who will remove the background or any additional object in the image. The OFIs produced by this second method are called manual OFIs. The manual method is more accurate than the automated API method and might lead to better results. The only difference between the second and the third models is the fact that the second model uses the automatic method while the third model uses the manual method. This is also the difference between the fourth and the fifth models.

In this paper, the five models are tested to answer the following questions:

- Will the model have a better validation accuracy when only the original set of images is used?
- Will it have better validation accuracy when only the set of OFIs is used?
- Will we have better validation accuracy when both sets are used?
- Are models with manual OFIs more accurate than those which use automatic OFIs?

## 2 RELATED WORK

In deep learning, augmentation is a commonly used practice since the 80's and early 90's (Simard et al., 1992). It is considered a critical component of many ML models (Ciresan et al., 2010; Krizhevsky et al., 2012; LeCun et al., 2015).

Augmentation is of paramount importance in extreme cases where only few training examples are available (Vinyals et al., 2016). In fact, it has been considered an extremely important element of numerous successful modern models. Among these models are: the AlexNet model (Krizhevsky et al., 2012), All-CNN model (Springenberg et al., 2014), and the ResNet model (He et al., 2016).

In some implementations, experts were able to rely on data augmentation, apply it heavily, and achieve successful results (Wu et al., 2015). This is not the case with computer vision only. Data augmentation was effective with other domains also like text categorization (Lu et al., 2006), speech recognition (Jaitly & Hinton, 2013), and music source separation (Uhlich et al., 2017).

When given sufficient data, deep neural networks were providing exceptional results. They have been studied, proven, and demonstrated in many domains (Gu et al., 2015), including: image classification (Krizhevsky et al., 2012; Huang et al., 2016), natural language processing (Gu et al., 2015), reinforcement learning (Mnih et al., 2015; Foerster et al., 2016; Silver et al., 2016; Gu et al., 2016; Van Hasselt et al., 2016), machine translation (Wu et al., 2016), synthesis (Wang et al., 2017), and many more. In all these experiments and implementations, used datasets have been significantly large.

Augmentation is a vital technique, not only for cases where datasets are small but also for any size of dataset. Indeed, even those models trained on enormous datasets such as Imagenet (Deng et al., 2009) could benefit from data augmentation.

Without data augmentation, deep neural networks may suffer from the lack of generalization (Perez & Wang, 2017) and adversarial vulnerability (Zhang et al., 2017). Although CNNs have led to substantial achievements in image-processing tasks and image classification (Zeiler and Fergus, 2014; Sermanet et al., 2014), CNNs with abundant parameters might over fit. This is due to the fact that they learn very detailed features of supplied training images that do not generalize to images unseen during training (Zeiler and Fergus, 2014; Zintgraf et al., 2017). Data augmentation has been considered a solution to this overfitting problem (Krizhevsky et al., 2012; He et al., 2016; DeVries and Taylor, 2017).

Nowadays, data augmentation techniques are gaining continuous attention (DeVries and Taylor, 2017; Zhong et al., 2017; Zhang et al., 2017). When it comes to image recognition, data augmentation is one of the fundamental building blocks for almost all state-of-the-art results (Ciresan et al., 2010; Dosovitskiy et al., 2016; Graham, 2014; Sajjadi et al., 2016). Nonetheless, because augmentation strategies may have a large impact on the model performance, they require extensive selection and tuning (Ratner et al., 2017).

In fact, data augmentation experts proposed cutout (DeVries and Taylor, 2017). At every training step, cutout randomly masks out a square region in an image. It is an extension of dropout.

Random erasing is similar to cutout (Zhong et al. 2017), as it masks out a subregion in an image but with the following three differences: i) It randomly chooses whether or not to mask out; ii) It uses a random size; and iii) It uses a random aspect ratio of the masked region.

On the other hand, Mixup $\alpha$-blends two images to form a new image (Zhang et al., 2017). Mixup also behaves like class label smoothing. It mixes the class labels of two images with the ratio $\alpha : 1 - \alpha$ (Szegedy et al., 2016).

Other researchers proposed techniques such as random-cropping and horizontal-flipping (Krizhevsky et al., 2012).

These useful augmentation methods have been applied to modern deep CNNs. They were proven to be very efficient and demonstrated the added value of data augmentation.

After ResNet was proposed (He et al., 2016), new network architectures have been proposed as well (Zagoruyko and Komodakis, 2016; Han et al., 2017). Pixel-dropping, however, injects noise into the image (Sietsma and Dow, 1991).

In short, there are numerous data augmentation techniques. Figure 1 summarizes them.

Each line with a shape in Figure 1 tells which data augmentation method is used by the corresponding meta-learning scheme. For instance, meta-learning using mixing images is covered in smart augmentation (Perez & Wang, 2017).

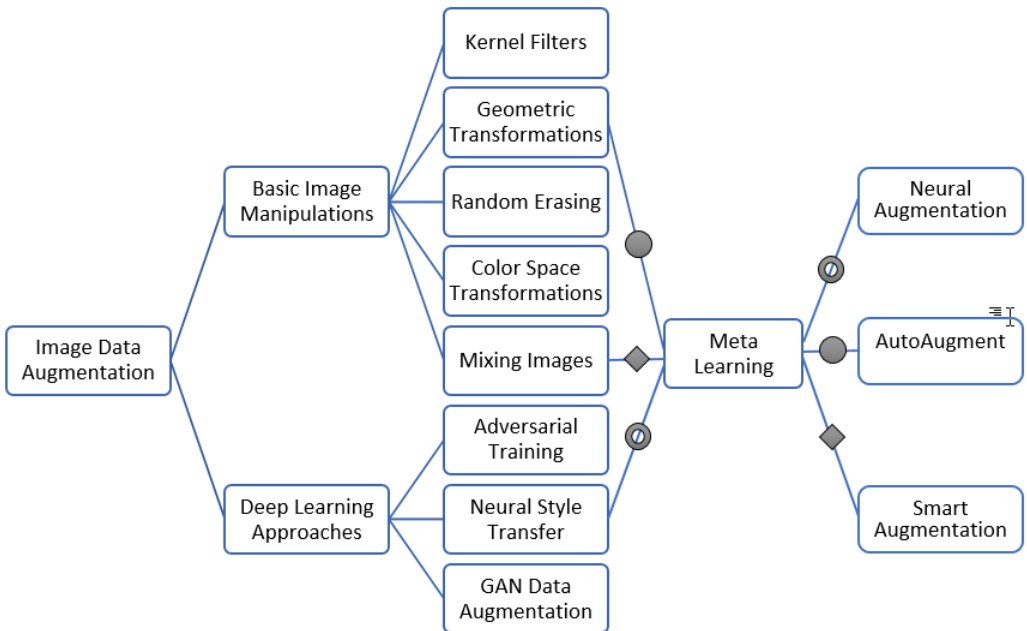

Figure 1 - A taxonomy of image data augmentations covered

## 3 THE OFI TECHNIQUE

Computer-vision engineers use images to train their models. These images include the object that needs to be correctly classified. There is, nevertheless, a small problem with these images: they include other objects that we do not want to train the model on.

If we want to train a model to classify images of dogs and cats, we should always ask ourselves the following questions:

- Why should we train the model on something that might not be seen later?
- Why, for example, should we train the model on a cat with a ball behind it, but when the training ends, we'll ask the model to classify an image without a ball?
- Once the model is trained, it will classify an image of a cat with a tree, a fence, or a couch behind it, so why train the model on a cat with a ball, while real-life images might include millions of other things?

It is a good practice to try to train a model using images that include the labeled object only and nothing else. This is what the OFI technique attempts to do. For every image in the training set, another image is generated where the labeled object is kept untouched and everything in the background is made transparent as shown in Figure 2.

When such an image is supplied as a part of the training set, the model is instructed to focus only on the labeled object. There will be no other objects for the network to be trained on.

Here, the scenario can be seen exactly as training a baby on what a cat is. When that baby sees only a cat in the image, s/he would not get confused which object is the cat. Otherwise, some questions need to be answered:

- What if there were many balls in the many images we have?
- Would our model get a little confused by those additional objects and try to be trained on?
- Would our model keep changing its parameters and weights based on the additional objects in the image?

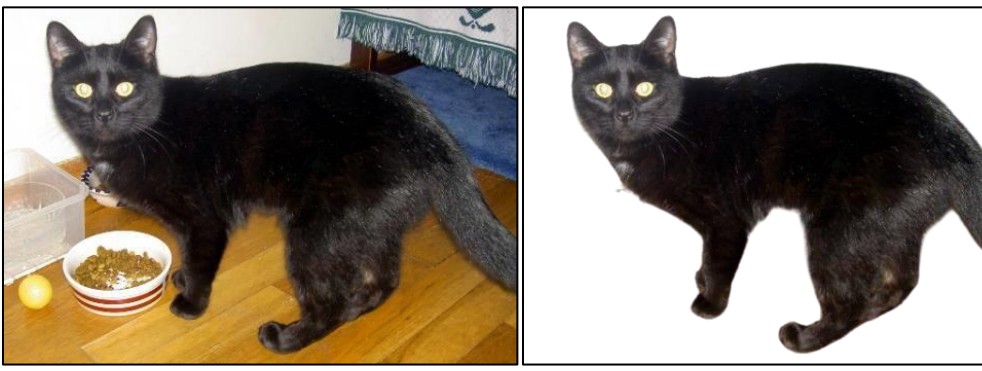

Figure 2 – Same image with and without the background

## 4 THE EXPERIMENTS

### 4.1 USED MODELS

With all the above questions in mind, several experiments need to be conducted to validate the OFI technique. The tested models are listed in Table 1. The goal behind checking all these models is to determine which model would produce the best results in terms of validation accuracy.

Table 1: Description of Models Used

| MODEL # | MODEL NAME | CONTENT OF THE TRAINING FOLDER |
|---|---|---|
| 1 | Original | This model has been trained using 1,000 images for cats and another 1,000 images for dogs. 

 These images include the labeled objects among other objects in the background, as it is done in the image to the left in Figure 2. 

 They are taken from the famous Kaggle Dogs vs Cats dataset. The training folder includes 2,000 images while the validation folder has 1,000 images only. |
| 2 | OFI Automated | This model includes a copy of the images used in model #1, but all backgrounds are erased using an online API called remove.bg. 

 The process of background removal is completely automated. All images in the training folder will be sent to the online API, in turn. The returned OFI image is then stored in the training folder. 

 Once all images are processed, the training can start. All the images will then include one object only, just like the image to the right in Figure 2. |

| MODEL # | MODEL NAME | CONTENT OF THE TRAINING FOLDER |
|---|---|---|
| 3 | OFI Manual | This model is similar to the previous one (model 2), but instead of using an API to edit the images and remove additional objects, a human photo editor will edit all the images manually.

The question is: will this slow process make the accuracy of this model higher than the accuracy of the previous automated model? |
| 4 | Mixed Automated | This model includes those images used in model #1 and those used in model #2. Will using both types of images result in a better model? |
| 5 | Mixed Manual | This model includes those images used in model #1 and those used in model #3. Which mix will result in a more accurate model: using original images with the manual OFIs or with the automated OFIs? |

4.2 NETWORK DESIGN

To test the above models, Tensorflow 2.3.0 has been used. The following is the design of the neural network:

```
model = tf.keras.models.Sequential([
   tf.keras.layers.Conv2D(32, (3,3), activation='relu', input_shape=(150, 150, 3)),
   tf.keras.layers.MaxPooling2D(2, 2), tf.keras.layers.Conv2D(64, (3,3), activation='relu'),
   tf.keras.layers.MaxPooling2D(2,2), tf.keras.layers.Conv2D(128, (3,3), activation='relu'),
   tf.keras.layers.MaxPooling2D(2,2), tf.keras.layers.Conv2D(128, (3,3), activation='relu'),
   tf.keras.layers.MaxPooling2D(2,2), tf.keras.layers.Dropout(0.5),
   tf.keras.layers.Flatten(), tf.keras.layers.Dense(512, activation='relu'),
   tf.keras.layers.Dense(1, activation='sigmoid')
])
```

The following settings are used for compilation:

```
model.compile(loss='binary_crossentropy', optimizer=RMSprop(lr=1e-4), metrics=['accuracy'])
```

Two image generators are used to rescale the images and do other image augmentation operations as follows:

```
train_datagen = ImageDataGenerator(
    rescale=1./255, rotation_range=40, width_shift_range=0.2, height_shift_range=0.2,
    shear_range=0.2, zoom_range=0.2, horizontal_flip=True, fill_mode='nearest')
test_datagen = ImageDataGenerator(rescale=1./255)
train_generator = train_datagen.flow_from_directory(
    train_dir, target_size=(150, 150), batch_size=20, class_mode='binary')
validation_generator = test_datagen.flow_from_directory(
    validation_dir, target_size=(150, 150), batch_size=20, class_mode='binary')
```

Finally, this is how the model is trained:
```
history = model.fit(
    train_generator, steps_per_epoch=100, epochs=150,
    validation_data=validation_generator, validation_steps=50, verbose=2)
```

All the above settings apply to all the models in Table 1. The only difference between a model and another is the content of the training folder. Accordingly, the steps per epoch might be changed based on the number of training images.

### 4.3 RESULTS

All the models in Table 1 were tested and the results are presented in Table 2.

The top row shows the traditional scenario, where the original images are used without any kind of editing. After 150 epochs, the validation accuracy is 80%. Will this accuracy be higher when the OFIs are used?

The second row in Table 2 includes the result of using OFIs only without using the original ones. These images are automatically generated by calling an API just before the training starts. The training accuracy is 88% while the validation accuracy is 66%. Why is there such a significant difference? This is because the model is trained on an object that exists in the image alone. It never expected to see a dog next to a tree, for example. It always expects to see a dog alone or a cat alone. In this case, the training data set is not similar to the validation data set. This is why we have a difference of around 22% between training and validation accuracy.

The third row in Table 2 shows the results of using OFIs that were manually produced. When a human photo editor keeps one object in the image and removes all other objects, the image tends to be more accurate. The API might sometimes crop a part of the labeled object or leave a part of another object. Thus, it is expected to produce a better result from training the model using manually edited images. Again, these images alone are misleading because the training data set is not similar to the validation data set. This is why there is a gap between training and validation in both accuracy and loss curves.

The fourth row in Table 2 represents the results after using the original 2,000 images along with the OFIs. This mix seems to be successful. It enables the model to receive training on two types of images. This is the same as giving more weight to the labeled object itself because it will be seen twice. The first time, the object will appear alone in the image and the second time, it will show along with other objects (or vice versa).

The difference between the last two rows is that the model in the fourth row uses an API to generate the OFIs, while the model in the fifth row uses OFIs that are manually generated by a human photo editor. The last model has a higher accuracy of 88% instead of 84% because it uses images edited by an expert, not by an API that might crop a part of the labeled object or leave a part of another object. A balance needs to be achieved between the pain needed to manually edit thousands of images and the gain of 4% in terms of accuracy.

A quick comparison between the first and the last rows of Table 2 would also lead us to the two following observations:

First, the OFI technique is just another successful approach that can be used to augment data and almost double the size of the image set. Indeed, some images cannot be automatically processed.

Second, using the OFI technique might be able to enhance the accuracy of the model from 80% (as in the top row of Table 2) to 88% (as in the last row).

### 5 CONCLUSIONS

When very limited data sets are available, sets of images can be augmented through the use of a mixed model that includes two sets of images (original images and OFIs). As shown in Table 2, the last model is 8% more accurate than the first one. It is recommended that developers of artificial intelligence (AI) engines include the OFI technique in their packages to enable users to use this feature on the fly the same way TensorFlow users, for example, use ImageDataGenerator to augment data.

Table 2: Results of the different models

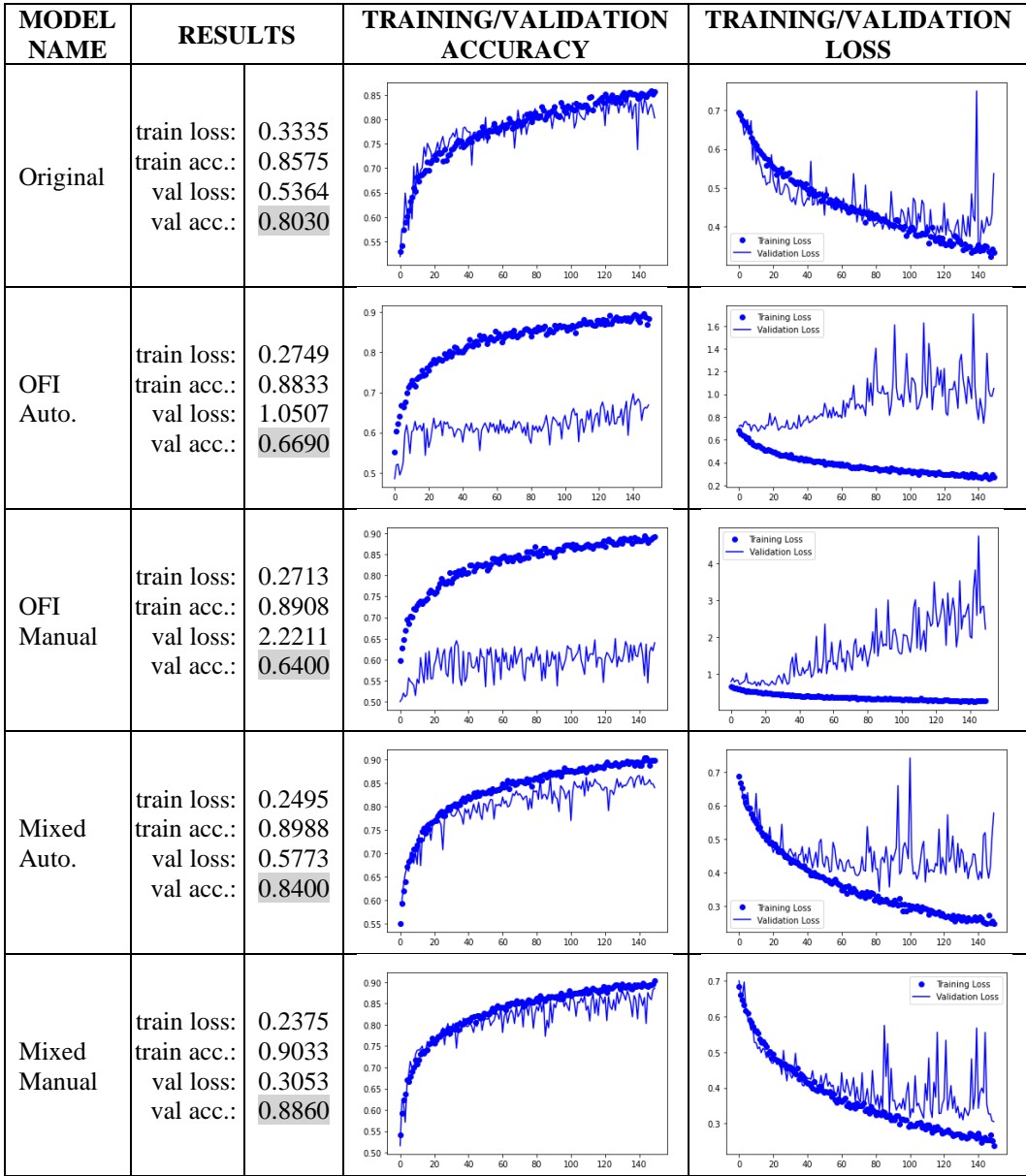

| MODEL NAME | RESULTS | | TRAINING/VALIDATION ACCURACY | TRAINING/VALIDATION LOSS |
|---|---|---|---|---|
| Original | train loss: train acc.: val loss: val acc.: | 0.3335 0.8575 0.5364 0.8030 | | |
| OFI Auto. | train loss: train acc.: val loss: val acc.: | 0.2749 0.8833 1.0507 0.6690 | | |
| OFI Manual | train loss: train acc.: val loss: val acc.: | 0.2713 0.8908 2.2211 0.6400 | | |
| Mixed Auto. | train loss: train acc.: val loss: val acc.: | 0.2495 0.8988 0.5773 0.8400 | | |
| Mixed Manual | train loss: train acc.: val loss: val acc.: | 0.2375 0.9033 0.3053 0.8860 | | |

## 6 FUTURE WORK

Two issues need to be tackled. The first is that an error might happen during the automated process when removing a background. This happens when the API cannot identify the background. Our tests show that other APIs can be used when the first one fails. During this research, the API (remove.bg) failed in removing the background for only 6% of images. The second issue happens when the labeled object (a dog, for example) is in the background. When the background is removed, the dog will be also removed, as shown in Figure 3. The question, then, would be: can we detect such specific scenario? Although this is not very common, it needs to be handled properly.

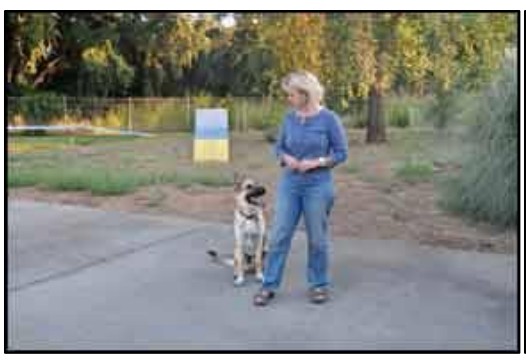 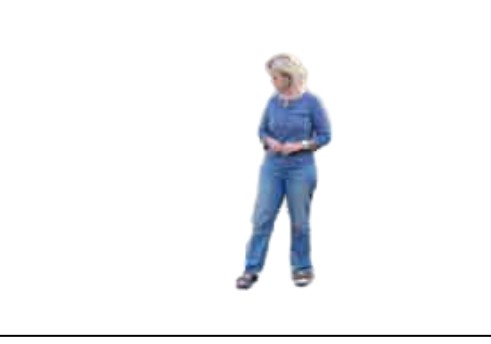

Figure 3 – When the background is removed, the dog vanishes (source: whole-dog-journal.com)

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
