# OpenReview forum: "USING OBJECT-FOCUSED IMAGES AS AN IMAGE AUGMENTATION TECHNIQUE TO IMPROVE THE ACCURACY OF IMAGE-CLASSIFICATION MODELS WHEN VERY LIMITED DATA SETS ARE AVAILABLE"
_ICLR.cc/2021/Conference — Reject_

### Official Review · AnonReviewer2 · 2020-10-26
**Recommendation to reject**

**Rating:** 3
**Confidence:** 4

**Review:**

Summary: Authors propose an augmentation technique for image classification. The augmented image is obtained by segmenting the salient object and masking the background. Therefore the technique gives one additional augmented image per each training image. The authors show an improved performance when using this augmentation on binary classification task (cats vs. dogs) using a dataset of 2000 images.

I find the paper in its current form lacking in many aspects, mainly: questionable novelty, bad presentation, minimal experiments.

Concerning novelty, the authors themselves remark: "It is a good practice to train a model using images that include the labeled object only". The section on related work is very superficial ignoring the achievements thanks to synthetic images. The approaches such as the following, used usually in the context of object detection, should be considered prior art and at least properly discussed, e.g.:

- Gupta et al, "Synthetic Data for Text Localisation in Natural Images", CVPR, 2016.
- Dwibedi et al, "Cut, Paste and Learn: Surprisingly Easy Synthesis for Instance Detection", ICCV, 2017.
- Georgakis et al, "Synthesizing Training Data for Object Detection in Indoor Scenes", RSS, 2017
- Dvonik et al, "Modeling Visual Context is Key to Augmenting Object Detection Datasets", ECCV, 2018
- Tripathi et al, "Learning to Generate Synthetic Data via Compositing", CVPR, 2019

There are many places in the manuscript where the sense of the text is not clear or the text appears out of context, e.g.: "After ResNet was proposed (He et al., 2016), new network architectures have been proposed as well (Zagoruyko and Komodakis, 2016; Han et al., 2017). Pixel-dropping, however, injects noise into the image (Sietsma and Dow, 1991)." The text seems in general unfinished and outdated at the same time (no citations of prior art newer than 2017).

Concerning experiments. The authors speak about lack of data but instead of evaluating on some few-shot datasets (e.g., mini-Imagenet or omniglot), they evaluate on a dataset with two classes and 1000 training images per class. The experiments are limited to evaluating a a model trained on the original unaugmented dataset, augmented using automatic segmentation, and augmented using manual segmentation. There is no comparison to other methods, not even the ones cited in the paper. Given obtaining segmentation can be costly I would expect the authors to test whether it is necessary at all and whether cutting the object to the bounding box is not enough. I would also expect a discussion of when can background removal actually hurt the performance (e.g., because it is predictive of the target class).

---

### Official Review · AnonReviewer4 · 2020-10-28
**Not acceptable**

**Rating:** 2
**Confidence:** 5

**Review:**

This work proposes to augment object focused image to improve image classification. Essentially, this work tries to remove background from original image using an existing algorithm and human editor, and train an image classification model using different combinations of original images and background-removed image. Five simple models are trained and their performance is compared to show that using background-removed images together with original images can help improve accuracy on validation set on a simple dataset.

Overall, this paper has serious flaws and is far from the level of ICLR. It is more like a course project report than a scientific paper. It lacks rigorous experiment design and result analysis, and the conclusion is not surprising or insightful. A clear rejection.

Detailed comments:

Technical:
- Although it makes sense to augment. input images to improve model performance, the design of the proposed work is less convincing. When talking about data augmentation, there usually won't be human editor/annotator involved and all the operations should be performed on-the-fly algorithmically. However, in this work, the background removal for original image is done either by a human editor or a third-party tool, which is not scalable at all.
- The technical part is also significantly limited. The background removal assumes that there is only one dominant object related to the label in an image. This simplifies the real problem too much. As the paper shows in Figure 3, this poses a serious problem when there are more than one object and the object of interest is not dominant. I do not think the proposed method can be applied to more complex scenarios at all.
- The paper states "When that baby sees only a cat in the image, s/he would not get confused which object is the cat." However, this may not be true. Actually we haven't really understood how exactly human's perception/neural system works.
- The performance of the model trained on images using the third-party background removal tool heavily depends on the quality of the tool. Since there is no evaluation on the tool, we do not know how the quality of background removal contributes to model performance improvement/degradation.

Experiments:
- The proposed model is only evaluated on one simple dataset containing only cats and dogs. It is more a sanity check/toy example than a realistic evaluation setting since there is just binary classification. Larger-scale datasets with multiple classes should be used.
- The model evaluated is a simple convolutional network, and not a standard model; i.e., ResNet, VGG or other widely used model, so it is difficult to say whether the conclusion drawn from this model can by applied to other models and datasets.
- Why not mix human edited images and images modified by the tool and train a model on all the images to see how it works?

---

### Official Review · AnonReviewer3 · 2020-10-29
**The paper proposes using object focused images for data augmentation in CNN training; it is of limited significance**

**Rating:** 5
**Confidence:** 5

**Review:**

The paper presents an object focused image based technique for CNN training, in which, for every image in the training set, another image is generated where the labeled object is kept untouched and everything in the background is rubbed out. A simple comparison is made among five scenarios on using separately or combined the original and OFI, illustrating that in the examined experiment somewhat better accuracy is obtained when combining them.

The presented research cannot be considered novel. Extracting regions of interest in images and using them for analysis, or extracting points of interest (e.g., facial landmarks) and using them together, or in addition, to the original images (e.g., faces) has been a frequently used approach to enhance object classification or regression tasks. The presented comparison has some interest, but it lacks even the required experimental validation, by comparing this data augmentation to any other competing approach - described in Figure 1.

In particular:

Section 2 on related work makes reference to a large number of techniques used for data augmentation, which are summarized in Figure 1. Nevertheless, none of these techniques is used to compare to the OFI based method.

The results presented in Section 4 are of low significance. Testing in scenarios 2 and 3 would be meaningful if the statistics were similar to training; thus, obtaining lower accuracy was expected. Similarly, when comparing scenarios 1 and 4, with the latter containing augmented data, it was expected to see a better performance in the latter case.

Various presentation and language errors should be corrected (missing references, poor language, etc).

---

### Official Review · AnonReviewer1 · 2020-10-30
**This paper does not step to the level to be accepted at ICLR.**

**Rating:** 3
**Confidence:** 5

**Review:**

The paper presents an image data set augmentation approach (4 variants+ original data set variant) in order to cater to the lack of sufficient data for learning purposes by increasing available image data sets through creation of  variations of existing images.
The variations are all about extracting recognizable objects in images (Object-Focused Images or  OFIs), manually and automatically,  and combination with original data set images.

Questions:
- “In fact, when the OFI technique is used, the number of the images supplied nearly doubles.”
	- Does not this depend on the type of images in the data set: multi-object images, no discernible object sets (textures, backgrounds etc.)?
        - Image given as example in Figure 2 is quite simplistic as the cat figure is clearly easily isolated. You Should state the type of application your model is intended for as you use for such a simplistic application.

- In 3. THE OFI TECHNIQUE:
	- You state “It is a good practice to try to train a model using images that include the labeled object only and nothing else.”.
        	- Does not this go against building models that will use real life images and therefore generalize well? Except may be in controlled setting like manufacturing lines.

- In 4. THE EXPERIMENTS /4.1 USED MODELS:
		- You conduct 5 different tests (or models as you call them): Original images, Automatic OFIs, Manual OFIs, Original + Automatic OFIs and Original + Manual OFIs.
	 	- Did not you try: Original + Automatic OFIs + Manual OFIs? Why not, you had the data.

Comments:
-	In  1 INTRODUCTION, you state:
	 	- " benefit further from using the mixed data set of old and new images”. Rather original data set images and derived/augmented images.
	 	- "The automated method uses an Application Programming Interface (API) to remove the background of the image and leave only the labeled object (a cat or a dog) in the foreground.” An API is an interface to remotely run an algorithm/call a method, not the actual method.
	 	- For consistency, since you deemed each experiment as a model, asking “Will the model have a better validation accuracy when only the original set of images is used?” is not consistent. Rather “will the model with only the original set of images …”
	 	- This paragraph seems out of place ” After ResNet was proposed (He et al., 2016), new network architectures have been proposed as well (Zagoruyko and Komodakis, 2016; Han et al., 2017).”.

- In 3, THE OFI TECHNIQUE:
	- Am perplexed by these questions:
	 	- Would our model get a little confused by those additional objects and try to be trained on?
	 	- Would our model keep changing its parameters and weights based on the additional objects in the image?
- In 4. THE EXPERIMENTS /4.1 USED MODELS:
	- “When a human photo editor keeps one object in the image and removes all other objects, the image tends to be more accurate.” Not sure, this phrasing is”accurate”.

---

### Decision · Program_Chairs · 2021-01-07
**Final Decision**

**Decision:**

Reject

**Comment:**

The paper introduces an augmentation technique that, given an image with a detected object, keeps the object and removes the background.

The reviewers expressed numerous valid concerns about the paper's novelty, the setting (assumption that there's a single object), the scalability of the approach and the experimental setup, including the baselines used.

The authors have not addressed these concerns.